# Modern Methods of Business Valuation—Case Study and New Concepts

**Ireneusz Miciuła** [1],*, **Marta Kadłubek** [2] **and Paweł Stępień** [1]

1   Faculty of Economics, Finance and Management, Department of Sustainable Finance and Capital Markets, University of Szczecin, 70-453 Szczecin, Poland; pawel.stepien@usz.edu.pl
2   Faculty of Management, Department of Logistics and International Management, Czestochowa University of Technology, 42-200 Czestochowa, Poland; marta.kadlubek@wz.pcz.pl
*   Correspondence: ireneusz.miciula@usz.edu.pl

**Abstract:** In the modern world, the terms enterprise value and valuation are of great importance. Knowledge about how much an enterprise is worth is of fundamental importance for both the owner of that company and investors when negotiating the price of an enterprise at the time of conducting a commercial transaction. The article presents the goals of the company's valuation and characteristic stages of the company's life at which such valuation is necessary. The article classifies the methods of enterprise valuation used today. On this basis, the valuation methodology is presented according to the MDI-R concept (Assets, Income, Intellectual Capital-Market), which in a broad spectrum measures the effectiveness of the company's operations and, in accordance with the current features of good valuation, aims to determine the fair value of the company. The purpose of the article is to demonstrate the need to improve the code of conduct and valuation standards. As part of the implementation of the objective, multi-faceted and complex valuation issues are presented, as well as factors that may distort the determination of fair value. The methodology of the study is based on inferences about the methodology of business valuation, and verification is based on practical examples, by which a hypothesis on the existence of critical elements of valuation is verified that allows the use of broad subjectivity in estimating the value of assets. At the same time, the factors that determine the possibility of the existence of too wide a subjectivity in estimating assets, which is in contradiction with the features of good valuation, are presented. The attempt is made to draw attention to the threats arising from modern business valuation methodologies and their challenges in the future. Additionally, this article offers the authors' proposed hybrid method MDI-R, which draws from existing solutions to improve their functionality and applicability.

**Keywords:** corporate finance; creating value; valuation of the company; management; financial market

## 1. Introduction

The term "enterprise" or "company" can be understood as a separate economic entity, an economic entity producing and selling on its own account goods and services with the goal to maximize profits. Modern enterprise financial management is about maximizing its value. An enterprise is a special form of investment. The owners, by investing in their own capital resources, expect to obtain certain benefits resulting from the multiplication of capital invested in this way, which leads directly to the increase in the value of the enterprise they own. Recognizing at the same time that the economic essence of ownership issues is closely related to issues of utility and the problem of the monetary value of the object of ownership, the issues relating to enterprise value, its specifics, various conditions, as well as methods and training procedures, are invariably important.

Business valuation is a complex process that requires the application of the vast knowledge of many fields of science, and there are many scientific and practical problems associated with this [1]. Despite the fact that a group of specialists has already done much in terms of efforts to standardize the valuation process, there are still many unsolved problems or controversial solutions adopted. Methods of the valuation of enterprises and their organized parts have not been regulated legally as strictly binding [2]. Also, there is no closed and complementary set of rules applicable to this process. The lack of uniform regulations is primarily due to the fact that it is not possible to fully codify a process that may relate to entities with different specificities, legal forms, assets or ownership structures. However, there are standards that allow for its partial structuring. Therefore, in many countries of the world, for many years, there have been standards for business valuation [3]. They relate to the methodology of valuation and the range of expertise that the valuator must have. They were developed by professional organizations that contain specialists in dealing with valuations. Experts are bound by codified procedures and standards of conduct, which guarantee the comparability of valuations and ease of their verification. Such a situation contributes to the security of business transactions.

The need for business valuation results from economic development. Along with the globalization of the economy, which is accompanied by an intense flow of capital to a growing number of countries, valuation becomes necessary to the sale, privatization, mergers and acquisitions or creation of joint ventures and many other processes relating to enterprises [4]. Determination of the final value of the entity is difficult due to the subjectivity of the concept of "value" itself. The problem is also the fact that business valuation is the combination of both theory and practice. It also depends on the capabilities of the business model used by the specific economic entity. It should be remembered, though, that the actual market value of the enterprise is very rarely exclusively determined by the assets taken into account in the balance sheet. The actual valuation is determined by a number of variable factors, such as the economic situation of the country, attractiveness of the market, the company's development strategy, human resources, the nature and manner of the use of assets owned [5]. Therefore, it can be stated that business valuation is the process of estimation of the price for assets and benefits achieved by the company as a result of their effective management. It is carried out in the moment, since the market is like a living organism and new information affecting the condition and operations of enterprises occurs over and over again.

Therefore, because of the complicated, constantly changing processes of business valuation, it is important to establish certain norms and legal standards. The International Valuation Standards Council developed a document including international norms in this field [6]. It contains the guidelines recommended in the process of assessment of the company's value, as well as appraisal reports and recommendations concerning their application. Obviously, these rules are not strictly binding, however, they constitute a set of good practices and guidelines specifying certain generally accepted principles, both ethical and methodological. This aims at the elimination of significant disparities in relation to the results of the valuation made, e.g., with respect to the assets of the same type. These rules may be also applicable in the case of court disputes concerning the outcome of the valuation, as well as doubts about the valuation for tax purposes. In the countries where a certain framework and standards for estimating the value of assets have been determined, there is separate certification of experts in the field of valuation.

The objective of the article is to demonstrate the need to improve the code of conduct and valuation standards. Within the framework of the accomplishment of the objective, the multidimensionality as well as the complex issues of valuation are presented along with the factors that could distort the establishment of fair value.

## 2. Methodology

The research methodology was based on a review of scientific literature and conclusions drawn from business valuation methodology. As part of this methodology a detailed analysis was performed of the subsequent stages of business valuation methods used in business practice. Also in the article

practical examples were verified. On their basis, the hypothesis about the existence of key valuation elements was verified, which allows the use of broad subjectivity when estimating the value of assets. The research methodology used demonstrated the need to improve the code of conduct and valuation standards. Additionally, as part of the critical analysis, factors that may distort the determination of fair value were presented. Therefore, the original concept of MDI-R was presented to improve practical valuation methods. At the same time, the necessity of further development of valuation methods and the search for objective methods of fair value measurement for the conducted business was shown, which require further detailed research based on the analysis of numerical data on practical examples.

## 3. Literature Review

Business valuation is a set of procedures, analyses and assessments leading to the estimation of the company's value in monetary units for the specific moment [7]. Contemporary realities of the market economy and the globalization process have determined that business valuation is of fundamental importance for economic processes. Contemporary management of corporate finance consists in maximizing its value. Business valuation is the process of estimating the price of assets (fixed and current assets, as well as different intangible assets and characteristics) and benefits achieved due to their effective management [8]. Generally, the objective of business valuation is always to facilitate strategic decision-making in terms of organization, shares or investments. Valuation enables the selection of both ownership and financial options in assets and liabilities. It is actually the opinion of the value prepared by specialized experts, analysts and valuators on the basis of the collected and properly utilized information about the company considered and the environment of its operations [9]. At the same time, it can be acknowledged that the company's value is the market measure of the effectiveness and efficiency of actions taken by the enterprise. Despite such a crucial function performed by business valuation in the economy, its specificity and essence pose many problems. These are related to a variety of conditions and numerous procedures and methods serving business valuation. Another component resulting in problems in the valuation of fair value is the growing importance of intellectual capital. This is due to global technological and organizational transformations, which have led to the knowledge-based economy. Intellectual capital, among others, consists of legal assets, technology and relationships with customers. At the same time, among the issues of intellectual capital, there are many ambiguous and various solutions for both theory and practice. Due to difficulties in the valuation of intangible and legal assets, which primarily determine the contemporary value of enterprises, in particular highly developed ones, in terms of technology, one deals with difficulties in achieving the so-called fair business valuation [10]. The methodology of the valuation of intangible and legal assets is subject to constant changes in search of a universal method. Therefore, at present, in the subject literature, the conclusion is that, in order to make the best possible valuation, it is necessary to valuate individual components affecting the value of the company with separate methods that best reflect the nature of their value.

The existence of many subjective factors affecting the valuation may lead to abuse, pressure and the desire to influence the experts' decisions, which result in the distortion of fair value. Therefore, in order to streamline business valuation, there is the need to develop a synthetic and universal, yet consistent, methodology for the valuation of basic parameters. This also requires the implementation of appropriate regulations or standards concerning the generally accepted methods of business valuation since, depending on the subjective choice of the method by the appraiser, significant differences in the final valuation may be observed, resulting in low values of the company [11]. Therefore, the objective should be to develop standards that will determine acceptable methods (patterns) for specific industries or cases. All practitioners carrying out business valuations must accept certain fundamental principles. The appropriate model for the estimation of the value of the economic entity should not only inform about the total value but also indicate the structure of the sources of its creation. Therefore, business valuation methods should take into account as many components of the company affecting its value as

possible. Nowadays, the valuation process is already moving in this direction, the example of which is a simultaneous use of, most frequently, asset-based, income-based and comparable company methods to determine the final value of the company. The use of several valuation methods in the course of the applied procedure provides an opportunity to make rational decisions as to the final value of the enterprise [12]. In this paper, the problem areas are analyzed and indicated, which can be useful when building Polish business valuation standards. Undoubtedly, standardization will lead to a reduction in a certain degree of subjectivity, present in valuations. Moreover, valuations made on the basis of the same requirements will become comparable and more easily verifiable in terms of their correctness.

Contemporary realities of the market economy along with the globalization process have caused business valuation to become of fundamental importance for economic processes [13]. Additionally, growing information needs have led to the development of numerous methods of valuation. The enterprise (company) is an economic entity producing and selling goods or services for its own account and at its own risk, an objective of which is to maximize profits. Contemporary management of corporate finance consists in maximizing its value. This is due to the fact that the enterprise, as a separate economic and legal entity, is a particular form of investment. Owners, investing their own capital resources in its economic activities, expect to obtain certain benefits, mainly to multiply the capital invested in this way, which directly leads to an increase in the value of the enterprise they own. At the same time, when recognizing that the economic essence of the issue of ownership is closely linked to the issues of usability and the problem of the monetary value of an object of property, the issue relating to the category of the company's value appears to be invariably important. Business valuation is the process of estimating the price of assets (fixed and current assets, as well as different intangible assets and characteristics) and benefits achieved due to their effective management [14].

The need for business valuation results from economic development. Along with the globalization of the economy, which is accompanied by an intense flow of capital to a growing number of countries, valuation becomes necessary to the sale, privatization, mergers and acquisitions or creation of joint ventures and many other processes relating to enterprises. It is also important for value management of subsidiaries located in the developing countries. Generally, the objective of business valuation is always to facilitate strategic decision-making in terms of organization, shares or investments. Valuation enables the selection of both ownership and financial options in assets and liabilities. It is actually the opinion concerning the value, prepared by specialized experts, analysts and valuators on the basis of the collected and properly utilized information about the company and the environment of its operations [15]. At the same time, it can be acknowledged that the company's value is the market measure of the effectiveness and efficiency of actions taken by the enterprise.

The process that aims to determine the value of the company is valuation. The term "enterprise valuation" means that the subject of the valuation is the economically and legally isolated organizational unit, with specific potential in the form of fixed and current assets, as well as different intangible assets and characteristics. Valuation can be treated as the opinion, judgment, estimation of the preciousness of something. According to Miles, valuation is an opinion concerning the value, usually made in writing and, at the same time, it is the process of estimating the value of the cost of the asset, a group of assets or all the assets belonging to the business or the specific investment [7]. However, despite such a crucial function performed by business valuation in the economy, its specificity and essence pose many problems as measures of effective operations, among others. This is related to a variety of conditions and numerous procedures and methods serving business valuation. At the same time, the reasons for the contemporary crisis of confidence in the methodology of business valuation are presented and the directions of future development are indicated, which is of fundamental importance for the operations and opportunities of the conducted business. The essence of the valuation of the company is to give its value as expressed in the specific monetary units using set prices, rules and analyses.

The research methodology is based on reasoning on the basis of the methodology of asset-based methods of business valuation and the analysis of practical examples in order to verify the hypothesis

about the existence of critical components of valuation that enable wide subjectivity in estimating the value of assets.

*The Process of Business Valuation–The Essence and Classification of Methods*

In the contemporary world, the value of the company and its valuation are of crucial importance. The value of the enterprise

- is an invariably important issue of the economic essence of ownership, which is closely linked to issues of usability and the problem of the monetary value of an object of property,
- is the market measure of the effectiveness and efficiency of actions taken by the enterprise.

The process of the valuation of the company, its specificity and essence pose many problems and controversies. This is connected with the essence of monetary valuation, which is a subjective measure. Then, there are various conditions and numerous procedures and methods for business valuation. The knowledge of the company's value is of great importance for making strategic decisions concerning characteristic stages of the company's operation [16]. The essence of the business valuation of the company is to give its value expressed in the specific monetary units using set prices, rules and analyses. Table 1 contains the objectives of business valuation under different conditions, which indicate fundamental importance for the conducted business.

**Table 1.** The objectives of business valuation.

| Internal | External | Internal–External |
|---|---|---|
| - Ability to control the capital invested by the owner to multiply value<br>- Measurement of the value of shares for the purposes of their presentation<br>- Acceptance of new shareholders or exclusion of some of the existing ones<br>- Change in the legal form of the business<br>- Management contracts, remuneration systems based on value creation<br>- Identification of value determinants<br>- Strategic planning<br>- Division of the company | - Dimension of taxes<br>- Determination of the amount of stamp duty, notarial fees, etc.<br>- Determination of the amount of insurance premiums<br>- Public offers<br>- Determination of the amount of compensation arising from insurance | - Purchase or sale of the company<br>- Ownership transformations<br>- Privatization and re-privatization<br>- Transfer of the company under the rent, franchise or lease<br>- Merger of enterprises<br>- Valuation of listed companies for the comparison with the stock market valuation<br>- Sale of newly issued shares<br>- Loan and credit collateral |

Source: Own study based on: [7,17].

Accurate preparation of the business valuation helps in achieving better trading terms since, in the course of negotiations, it allows for relying on facts and not intuition, feelings and emotions [18]. At the same time, it enables avoiding the situation where the owner's idea significantly exceeds the actual value of the company or, on the contrary, where this idea definitely does not estimate the company's value. Therefore, the situation in which the transaction under objective conditions might fail is avoided. There are five functions of valuation resulting from the reasons for carrying it out [19]:

• Advisory (decision-making) function

The essence of the advisory function (also known as decision-making) is to provide the necessary information on the value of the company in relation to the intended execution of certain transactions.

• Argumentative (justifying) function

The implementation of this function consists in the skillful use of information obtained in the course of valuation. This is about the selection of information that will strengthen the bargaining power of the party to the transaction.

• Mediation function

The mediation function, also known as negotiating, refers to situations where the opinions of parties to the transaction as to the value are significantly divergent.

• Security function

Its essence is to provide information on the value of the company for the purposes of protection against the adverse effects of disputes arising in connection with the value.

• Information function

Its essence is to provide information obtained in the process of valuation for the purposes of enterprise management. The recipients of this information are investors, banks, trading partners, customers, financial analysts, authorities at different levels, etc.

After conducting the preliminary analysis of the subject and purpose of the business valuation, it is necessary to choose the method that is the most adequate to the situation of the enterprise and the specificity of its industry. Contemporary realities of the market economy and the globalization process have determined that business valuation is of fundamental importance for economic processes [5]. Additionally, growing information needs have led to the development of numerous methods of valuation. The determinants of the selection of the business valuation method include [20]:

- Valuation objective;
- Who orders (recipient);
- Type of the company due to usability;
- Economic condition of the company and the condition of the environment (economy, industry, region);
- Type, scale and diversity of business;
- Type and number of assets;
- Operation and development prospects of the company;
- Type and quality of information about the company and the market that it is possible to obtain;
- Approaches and types of value in business valuation.

A wide range of practitioners carry out business valuation on a daily basis. Therefore, valuation should be perceived as a practical activity and defined as a way of value (monetary) measurement of the enterprise, i.e., its resources and economic effects of decisions taken.

Fair value is the amount for which an asset can be exchanged if the transaction takes place under market conditions between interested parties who are not related to each other and possess the information that allows for full assessment of the value of the subject of the transaction. At the same time, business valuation is a complex process that is able to illustrate the actual and fair value of the company only if it is carried out in accordance with the so-called characteristics of good (reliable) valuation, which include [21]:

- Compliance of the valuation with the facts;
- Timeliness of data, transparency and relative simplicity;
- Clearly defined purpose of its preparation;
- Being based on the financial data of the company;
- Not being made exclusively on the basis of the value of the company's assets unless it concerns the so-called liquidation method;
- Taking into account income and intangible factors;
- Taking into account the company's development forecasts and risk factors;
- Taking into account all relevant information which affects the valuation and is available in the process of its preparation;
- Being objective and reliable.

On the other hand, the selection of the valuation method itself constitutes the most important part of the process of estimating the actual fair value of the company and must be adjusted to the subject of the valuation. Despite the fact that a wide range of practitioners carry out business valuation on a daily basis, this process still requires improvement. Table 2 presents the methods of business valuation the most frequently applied in practice.

**Table 2.** The classification of business valuation methods.

| Business Valuation Methods | | | | |
|---|---|---|---|---|
| **Asset-Based** | **Income-Based** | **Mixed** | **Comparable Company** | **Unconven-tional** |
| - Book value method<br>- Adjusted net assets method<br>- Replacement method<br>- Liquidation method | - Discounted dividend method<br>- Discounted cash flow method<br>- Discounted future earnings method | - Average cost method<br>- Swiss method<br>- Berlin method<br>- Excess earnings method<br>- Stuttgart method<br>- UEC[1] method | - Multiples method<br>- Method of comparable transactions | - Option theory-based methods<br>- Time lag methods<br>- Others |

[1] The name of the method comes from the commission called by Union Europeene des Experts Comptables Economiques et Financiers. Source: Own study based on: [22].

Another component resulting in problems in the valuation of fair value is the growing importance of intellectual capital. This is due to global technological and organizational transformations that led to the knowledge-based economy [23]. Intellectual capital, among others, consists of legal assets, technology and relationships with customers, etc. At the same time, among the issues of intellectual capital, there are many ambiguous and various solutions for both theory and practice. Figure 1 illustrates the changes in the significance of assets of enterprises for their valuation.

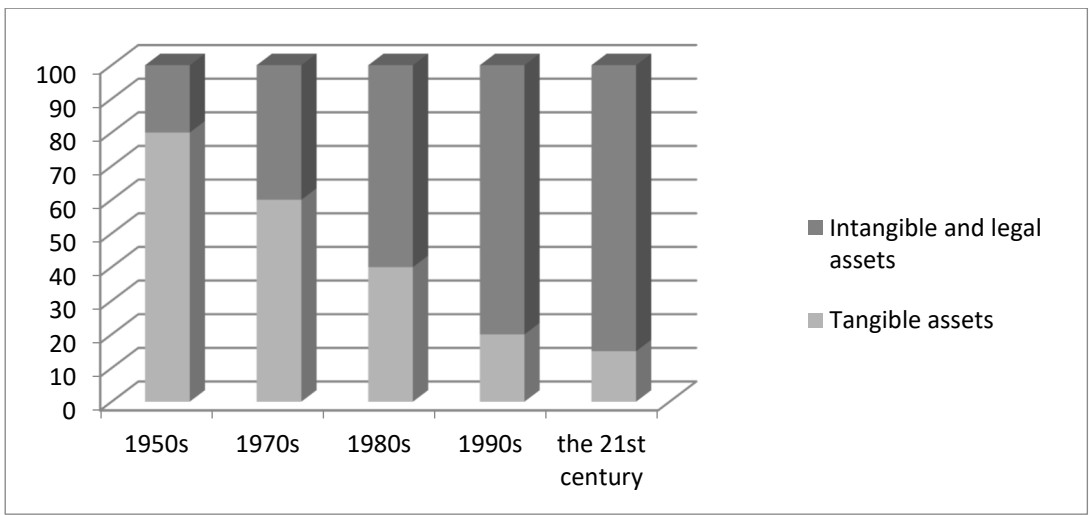

**Figure 1.** The significance of the values of tangible and intangible assets over the years. Source: Own study based on the data: [24].

Therefore, due to difficulties in the valuation of intangible and legal assets, which primarily determine the contemporary value of enterprises, in particular, highly developed ones, in terms of technology, one deals with difficulties in achieving the so-called fair business valuation. The methodology of the valuation of intangible and legal assets is subject to constant changes in search of a universal method. Therefore, along with the aforementioned components, these are the main reasons for the contemporary crisis of confidence in the methodology of business valuation. Therefore, at present, in the subject literature, the conclusion is that in order to make the best possible valuation it is necessary to valuate individual components affecting the value of the company with separate methods, which best reflect the nature of their value.

## 4. Results

An extremely important factor in the business valuation process is the appropriate selection of methods. This choice is determined not only by the objective of valuation and the situation of the valuated entity but also by the nature of the business and the specificity of areas of its business activity. The economic situation of enterprises, i.e., their market position, status of assets, ability to generate income, are some of the determinants having an impact on the selection of the valuation method. Additionally, in the course of analyses of information concerning the valuated entity and also within the framework of the application of the same method, one deals with the so-called critical points of valuation, i.e., components subjected to the subjective selection. The proper determination of the financial condition of the company can also be a source of differences, depending on the person carrying out the valuation and availability of information [25]. Additionally, it should be noted that each industry is characterized by a certain specificity, which has a large impact on many components that determine the valuation process. The existence of many subjective factors affecting the valuation may lead to abuse, pressure and the desire to influence the experts' decisions, which results in the distortion of fair value. Therefore, the following should be mentioned as risks and problems to solve in the future:

- Freedom of selection of input data,
- Use of wide subjectivity in the common valuation procedure,
- Subjectivity of selection of valuation methods and internal parameters,
- Lack of coherence in estimation of parameters,
- Lack of legal regulations and standards of valuation.

Therefore, in order to streamline business valuation, there is a need to develop a synthetic and universal yet consistent methodology for the valuation of basic parameters. This also requires the implementation of appropriate regulations or standards concerning the generally accepted methods of business valuation since, depending on the subjective choice of the method by the appraiser, significant differences in the final valuation may be observed with low values of the company [26]. The related works appeared due to the association of a group of specialists and practitioners. One of the tangible results of efforts to eliminate a number of risks is the announced *New Interpretative Note No 5—General Principles of Business Valuation* [27]. Despite a large number of ways of carrying out valuations and with the high quality of analytical work at the valuation, it should be remembered that a business is worth as much as someone is willing to pay for it. On the other hand, good and factual valuation is essential for preparing oneself for substantive discussions with the buyer. Within the framework of this paper, the essence, objectives and functions of business valuation were determined and the overall classification of valuation methods applied in the practice of economic life was made.

*4.1. The Examples of Business Valuation Using the Adjusted Net Assets Method–Case Study*

4.1.1. The Practice Example No. 1—Valuations Using the Adjusted Net Assets Method

Under ideal conditions, business valuation would consist in estimating the value of each asset individually and then subtracting all liabilities. The net asset value is then received, i.e., the value less liabilities, adjusted for the book value. For example, such a procedure occurs when valuating enterprises under court cases for division of assets or repayment of some of shares, which is illustrated by the example in the table below (Table 3).

**Table 3.** The listing of the values of fixed assets adjusted for the market values in relation to the book values.

| No. | The Name of the Fixed Asset | Book Value [PLN] | Market Value on the Valuation Date [PLN] |
|:---:|:---|:---:|:---:|
| 1. | A fiscal printer—Viking | 1.499 | 800 |
| 2. | A printer—Canon 250 | 0 | 210 |
| 3. | A computer set | 3.200 | 1.200 |
| 4. | A car—Toyota Avensis | 14.500 | 9.500 |
| 5. | Cell phones—Motorola M3588—2 pieces | 1.350 | 250 |
| 6. | A fax machine | 0 | 200 |
| 7. | Software—WF-MAG | 658.25 | 658.25 |
| 8. | A computer upgrade—HDD 4GB | 340 | 0 |
| 9. | A truck—Citroen Berlingo OP15937 | 15.100 | 12.000 |
| 10. | Etc. till the inclusion of all the assets | … | … |

Source: Own study based on the opinion of the court expert on behalf of the District Court of Katowice—Wschód in Katowice.

However, for practical reasons, only the most important balance sheet items are often subjected to adjustment, which are also different depending on the valuation objective. Due to the fact that the starting point of the valuation of the company is its net assets, there are many drawbacks to this method, which, to an extent dependent on the resources designed for valuation (mainly cash and time), are adjusted in the course of the valuation. The largest discrepancies between actual and balance sheet values are mostly caused by referring the balance sheet valuation to historical costs and the difficulty in clear classification of some items as equity or foreign capital (reserves, special funds, accruals). Therefore, in order to obtain the real value of the company, it is necessary to make adjustments to balance sheet items. When making adjustments, one should take into account the valuation objective. Other adjustments will be made when the company is purchased for resale, merger or significant reorganization. Adjustments of individual items are also dependent on the type of the conducted activity. The significance of balance sheet items is also assessed in order to reduce the costs of adjustments of less important items. Valuation should start with the determination if all the items are properly reflected in the balance sheet and if the items included in the balance sheet are not just empty records. Obviously, the balance sheet that is the basis for the valuation should be up to date.

The first item is intangible and legal assets. They are usually valuated under the market value; however, when they are unsellable they are valuated at zero. However, there are exceptions, e.g., non-transferable software licenses. In the case of business continuation, they will present measurable value. It should be taken into account that most development costs cannot be activated. Therefore, e.g., in research companies, the relevant adjustment may be essential. Another group is tangible fixed assets. If there is a secondary market for tangible fixed assets, the market price should be taken into account, since depreciation write-offs illustrate formal and not actual consumption of these assets. It is also possible to consider the price of a new fixed asset, taking into account a range of adjustments (due to technological underdevelopment, wear, expert opinion, market conditions, liquidity of the secondary market, etc.). It is also important to include, in the valuation, the fact that market prices may significantly vary depending on the stage of the business cycle [28]. The possessed land is usually underestimated, and making its value realistic results in an increased value of the company valuated. In the case of the right to perpetual usufruct over land, under the item of tangible fixed assets the difference can be found between the first (higher) and subsequent (lower) charges. This difference will be subjected to depreciation write-offs in accordance with the general rules. Therefore, the possession of the right for perpetual usufruct over land will increase the value of the company valuated.

When valuating receivables, despite the existing reserves, lowering adjustments are additionally made, which result from insufficient reserves. It is also necessary to indicate receivables that are not due to specific, already conducted transactions but agreements that cannot be included in the balance sheet on the day of its production [29]. It should be verified if the stocks included in the balance sheet actually exist and if the way of their valuation corresponds to the market value. It is also essential to determine whether the stocks owned by the company are not obsolete, broken or damaged. In the case of receivables, first of all, it should be checked if they are not underestimated and if there are not off-balance sheet obligations and what is the probability of their maturity. Such liabilities will additionally decrease the value of the company valuated. Also, tax liabilities may require adjustments.

While valuating financial assets, the market price of shares and/or stocks should be taken into account. Valuation ought to include the valuation of units in which the specific company has shares/stocks, as long as their value justifies such a way of conduct. When valuating cash, one should pay attention to whether it is cash or whether, under this item, there are also promissory notes and cheques that are associated with risk, which should be considered in appropriate adjustments. Prepayments and accrued income may also require adjustments. Most of all, it is necessary to determine if their allocation over time is correct. In the case of inappropriate estimation, accruals and deferred income may also require adjustments. A typical example would be not taking into account delayed invoices for external services that increase the costs of the specific period. Additionally, when business valuation is made with the intention to merge two entities, there are adjustments to harmonize the accounting approach used in each of the merged enterprises. Also, the need to pay dividends in the future or to recapitalize one of the merging enterprises by the third party will require adjustments increasing or decreasing the entity valuation.

Despite the fact that valuation is carried out on the basis of data resulting from the balance sheet, legal regulations or concluded agreements, this method is also not without a certain dose of subjectivity, since the valuator makes the decision on what adjustments and in relation to what assets they will be applied. Each business valuation should be approached individually, since it may turn out that each item in the balance sheet may require adjustments adapting it to the market value. The adjusted net assets method, due to less complicated assumptions necessary for its conduct and adjustment of prices to current net prices, is the most frequently applied asset-based method when making business valuations.

4.1.2. The Practice Example No. 2

The objective of the valuation in the Example 2 (Table 4) is to inform about the value of equity for the previous owner, who is interested in selling his or her block of shares and is planning to use the valuation as a starting point in negotiations.

Downward adjustments adopted in the valuation related to the following assets:

- Intangible and legal assets—this is an integral part of computer hardware. In this case, the equipment along with the software was included under the item of technical equipment and machinery;
- Means of transport—the market value of the means of transport was lower than their balance sheet value by 11.7%. One of the three vehicles had had an accident; therefore, its value was reduced, which resulted in a lower value of the whole item;
- Fixed assets under construction—these were repair expenditures in the company's main office; the liquidation value of 0 PLN was accepted for the balance sheet after adjustment;
- Long-term receivables—20% was assumed, considering the failure in getting back all the deposits;
- Long-term investments—revaluation of the B2X Ltd. company, in which the Service company has 51% of shares, lowered the value by 24.5% from the balance sheet value;
- Goods and receivables on account of supplies and services—the adjustment indicator of 20% was adopted.

Upward adjustments on the asset side are the following components:

- Real estate (buildings, premises)—revaluation prepared by valuators increased the value by 23.7%, to PLN 5125 thousand and this amount was accepted for the adjusted balance sheet;
- Technical equipment and machinery—were subjected to adjustment on the basis of current market prices and the total value increased;
- Other fixed assets—the appraisal reports produced by valuators indicated a higher value than the balance sheet value.

**Table 4.** The assets of the Service S.A. company prior to and after adjustments.

| | (PLN Thousand) | Prior to Adjustment | Adjustment % | Adjustment | After Adjustment |
|---|---|---|---|---|---|
| A. | **Fixed Assets** | 9453.39 | | | 10,303.05 |
| I. | **Intangible and Legal Assets** | 81.54 | | | 0 |
| 1. | Other intangible and legal assets (software) | 81.54 | | −81.54 | 0 |
| II. | **Tangible Fixed Assets** | 6550.52 | | | 7825.00 |
| 1. | Fixed assets | 6539.09 | | | 7825.00 |
| a) | Buildings, premises and civil engineering facilities | 4144.62 | 23.7% | 980.38 | 5125.00 |
| b) | Technical equipment and machinery | 1076.26 | 23.7% | 255.29 | 1331.54 |
| c) | Means of transport | 169.79 | −11.7% | −19.79 | 150.00 |
| d) | Other fixed assets | 1148.42 | 13.2% | 151.58 | 1300.00 |
| 2. | Fixed assets under construction | 11.43 | −100.0% | −11.43 | 0.00 |
| III. | **Long-term Receivables** | 583.01 | −20.0% | −116.60 | 466.41 |
| IV. | **Long-terms Investments** | 1258.22 | | | 950.00 |
| 1. | Long-term financial assets | 1258.22 | | | 950.00 |
| a) | in affiliated entities | | | | |
| | Shares in B2X Ltd. | 1258.22 | −24.5% | −308.22 | 950.00 |
| V. | **Long-term Accruals** | 980.10 | | | 980.10 |
| 1. | Deferred tax assets | 964.47 | | | 964.47 |
| 2. | Other accruals | 15.63 | | | 15.63 |
| B. | **Current Assets** | 3939.26 | | | 3692.45 |
| I. | **Stocks** | 1234.05 | | | 987.24 |
| 1. | Goods | 1234.05 | −20.0% | −246.81 | 987.24 |
| II. | **Short-term Receivables** | 461.54 | | | 390.26 |
| 2. | Receivables for the other entities | 461.54 | | | 390.26 |
| a) | On account of supplies and services, in the repayment period of up to 12 months | 356.39 | −20.0% | −71.28 | 285.11 |
| b) | Due to taxes, subsidies, custom duties, social and health insurances and other benefits | 79.60 | | | 79.60 |
| c) | Others | 25.55 | | | 25.55 |
| III. | **Short-term Investments** | 2000.00 | | | 2000.00 |
| 1. | Short-term financial assets | 2000.00 | | | 2000.00 |
| a) | Cash and other monetary assets | 2000.00 | | | 2000.00 |
| | - cash in hand and at bank | 1000.00 | | | 1000.00 |
| | - other cash | 900.00 | | | 900,00 |
| | - other monetary assets | 100.00 | | | 100.00 |
| IV. | **Short-term Accruals** | 243.68 | | | 243.68 |
| | **Total assets** | 13,392.65 | | | 13,924.22 |

Source: [30].

No adjustments were made to foreign liabilities; therefore, their balance sheet value was assumed in the amount of PLN 7980.28 thousand. The adjusted assets in the amount of PLN 13,924.22 thousand—the adjusted liabilities in the amount of PLN 7980.28 thousand equals the value of PLN 5943.95 thousand. A lot of professionals in the field of business valuation usually end the valuation with the adjusted net assets method at this stage. However, within the framework of the development of the method, it is worth taking into account other components, which may have an impact on the value of the company (value sources), e.g., trademark (brand), know-how, own patents or licenses. In the business practice, the adjusted net assets method usually indicates a lower value of the operating enterprise since it does not include, for example, the value of the company's contacts or knowledge of employees. Therefore, in the process of business valuation, the value determined by the asset-based valuation method was considered as the minimum value for the seller, which constitutes the company's assets.

### 4.1.3. The Practice Example No. 3

The ABC company, among its assets, has:

- The office building worth (according to the appraisal report of 2009) PLN 3 million,
- The production building worth (according to the appraisal report of 2009) PLN 5 million,
- The assembly line of 2003 worth (in the valuator's opinion) PLN 0.5 million,
- Stocks of materials and products worth PLN 4 million,
- Receivables worth PLN 3 million, PLN 0.5 million of which is uncollectible.

The total value of assets amounts to PLN 15 million (the market value of assets, which may significantly vary from the book value). At the same time, the value of liabilities of the company is two investment loans for a total value of PLN 8 million and liabilities to suppliers worth PLN 3 million. The value of liabilities amounts to a total of PLN 11 million. On the basis of the above, the adjusted net assets value amounts to PLN 15 − 11 = 4 million.

The advantages of the valuation using the adjusted net assets method include:

- Objectivity and ease in carrying out by oneself,
- Access only to basic data,
- Taking into account the condition and usability of assets for operation,
- Possibility to compare with the value determined using other methods,
- Possibility to determine the lower range of values in negotiations.
- On the other hand, the primary disadvantages include:
- Not taking into account important components of the company's value not recorded in the balance sheet, e.g., contracts of the company, knowledge of employees, possessed brands and value of trademarks;
- Possibility to determine only the value of assets in the categories of the so-called material substance, which usually underestimates the value of the operating enterprise.

Undoubtedly, the adjusted net assets method offers the greatest usability in the case of enterprises with a high share of fixed assets in the value of the whole company, i.e., traditional production companies. This is confirmed by the evolutionary history of business valuation methods. At the same time, despite the increasing importance of income-based and comparable company methods, nowadays asset-based methods are still the basis for the estimation of business value.

### 4.1.4. The Practice Example No. 4

Practical Example number 4 (Table 5) shows the valuation of the enterprise using the income method. This is one of the most popular methods of this type used in practice, namely, Discounted Cash Flows (DCF). For the calculation, the available econometric software used in business valuation practice in the market was applied.

**Table 5.** Example of valuation with the method of income (Discounted Cash Flows).

| XYZ S.A. | 2017 | 2018 | 2019 | 2020 | 2021 | 2022 | 2023 | 2024 | 2025 | 2026 | 2026+ |
|---|---|---|---|---|---|---|---|---|---|---|---|
| **Model DCF (Discounted Cash Flows)** | | | Forecast | Forecast | Forecast | Forecast | Forecast | Forecast | Forecast | Forecast | Forecast |
| 1. Operating result (EBIT) | 1531 | 1607 | 1688.1 | 1772.5 | 1861.1 | 1861 | 1861 | 1861 | 1861 | 1861 | 1861 |
| 2. Tax rate % | 19% | 19% | 19% | 19% | 19% | 19% | 19% | 19% | 19% | 19% | 19% |
| 3. Tax on EBIT | 290.9 | 305.5 | 320.7 | 336.8 | 353.6 | 353.6 | 353.6 | 353.6 | 353.6 | 353.6 | 353.6 |
| 4. Tax-adjusted operating result (NOPLAT) | 1240 | 1302 | 1367.4 | 1435.7 | 1507.5 | 1507 | 1507 | 1507 | 1507 | 1507 | 1507 |
| 5. Depreciation | 1317 | 1445 | 1597.6 | 1764.9 | 1949 | 2154 | 2384 | 2643 | 2936 | 3268 | 3644 |
| 6. Investment outlays (CAPEX) | 1310 | 1517 | 1646,5 | 1788.6 | 1945.5 | 2154 | 2384 | 2643 | 2936 | 3268 | 3644 |
| 7. Change in working capital | 180.0 | 171.8 | 180.3 | 189.4 | 198.8 | 0.0 | 0.0 | 0.0 | 0.0 | 0.0 | 0.0 |
| **8. FCF–free cash flow** | **1067** | **1058** | **1138.1** | **1222.7** | **1312.8** | **1507** | **1507** | **1507** | **1507** | **1507** | **1507** |
| 9. Risk-free rate% | 5% | 5% | 5% | 5% | 5% | 5% | 5% | 5% | 5% | | |
| 10. Beta indicator | 1.2 | 1.2 | 1.2 | 1.2 | 1.2 | 1.2 | 1.2 | 1.2 | 1.2 | 1.2 | |
| 11. Market premium % | 2% | 2% | 2% | 2% | 2% | 2% | 2% | 2% | 2% | 2% | |
| 12. Cost of equity % | 7.4% | 7.4% | 7.4% | 7.4% | 7.4% | 7.4% | 7.4% | 7.4% | 7.4% | 7.4% | |
| 13. Cost of debt % | 8% | 8% | 8% | 8% | 8% | 8% | 8% | 8% | 8% | 8% | |
| 14. Tax rate % | 19% | 19% | 19% | 19% | 19% | 19% | 19% | 19% | 19% | 19% | 19% |
| 15. Cost of debt after tax % | 6.5% | 6.5% | 6.5% | 6.5% | 6.5% | 6.5% | 6.5% | 6.5% | 6.5% | 6.5% | |
| 16. Value of equity (resulting from the valuation) | 14092.6 | 14092.6 | 14092.6 | 14092.6 | 14092.6 | 14092.6 | 14092.6 | 14092.6 | 14092.6 | 14092.6 | |
| 17. Value of debt | 7075 | 7075 | 7075 | 7075 | 7075 | 7075 | 7075 | 7075 | 7075 | 7075 | |
| 18. Share of equity | 66.6% | 66.6% | 66.6% | 66.6% | 66.6% | 66.6% | 66.6% | 66.6% | 66.6% | 66.6% | |
| 19. Share of debt | 33.4% | 33.4% | 33.4% | 33.4% | 33.4% | 33.4% | 33.4% | 33.4% | 33.4% | 33.4% | |
| ***20. Weighted Cost of Capital (WACC)*** | **7.09%** | **7.09%** | **7.09%** | **7.09%** | **7.09%** | **7.09%** | **7.09%** | **7.09%** | **7.09%** | **7.09%** | **7.09%** |
| 21. Discount indicator | 0.933 | 0.871 | 0.8142 | 0.7603 | 0.709 | 0.662 | 0.619 | 0.578 | 0.539 | 0.504 | 0.504 |
| 22. FCF growth rate after 2026 | | 0% | 0% | 0% | 0% | 0% | 0% | 0% | 0% | 0% | 0% |
| 23. Residual value after 2026 | | - | - | - | - | - | - | - | - | - | 21,255 |
| **24. Discounted FCF–Free Cash Flow** | **996.9** | **923.2** | **926.6** | **929.6** | **932.0** | **999.3** | **933.1** | **871.3** | **813.6** | **759.8** | **10,712** |
| **25. Discounted Free Cash Flow Ascending** | **996.9** | **1920.1** | **2846.7** | **3776.3** | **4708** | **5707** | **6640** | **7512** | **8325** | **9085** | **19,797** |

| | | |
|---|---|---|
| **1. Value of the Company from the Valuation** | 19,797.6 | |
| 2. Net debt at the end of 2018 | 5705.0 | |
| **3. Value of Equity from the Valuation** | **14,092.6** | |
| 4. Number of shares in the company | 1000 | |
| **5. Value of 1 share** | **14.09** | |

Source: [31].

The most common methods currently used in valuation practice are the adjusted net asset method and the discounted cash flow method [32]. They form the basis for determining the explanatory variables regarding assets and income options. To combine the advantages of both methods, mixed methods were created that look for optimal structural parameters.

*4.2. The Analysis of Methods and the Valuation Process to Establish the Determinants of Value Subjectivity*

Asset-based methods are historically the oldest concept of business valuation, adopting assets as the bases for determining the value. Therefore, the value of enterprises, being the effect of this valuation, is known as the asset value. This means that the company is worth as much as its valuated

assets. However, due to the fact that many companies are in debt, one may talk about the gross and net assets value, i.e., the value reduced by the value of debt. In these methods, the market value of the company, understood as the sum of the value of business asset liquidation, is subjected to valuation. The first and simultaneously the simplest valuation of assets is carried out using the net book value method (recordkeeping). The information included in the balance sheet is directly used. The formula for business valuation using the recordkeeping net assets value method:

$$WP = A - Po = KW,$$

WP—business value (net book value),
A—total balance sheet value of assets,
$P_o$—balance sheet value of foreign liabilities,
KW—balance sheet value of equity.

However, the book value of assets and liabilities does not usually equal their market value, which, in the current rapidly changing market conditions—in particular, in the segment of high technologies—may result in significant differences in value, which become unacceptable in order to determine the current fair value. The valuation based on the market value of the possessed assets and liabilities is known as the adjusted net assets method. The formula for business valuation using the adjusted net assets value [33]:

$$WP = AW - POW = KWW$$

AW—total adjusted assets value,
POW—value of adjusted foreign liabilities,
KWW—value of adjusted equity.

It is the most common method of valuation of business assets applied nowadays.

The objective of the replacement method is to estimate the total financial outlay that would be needed to restore individual assets of the valuated company. This method is often used by entrepreneurs making a decision on whether it is more profitable to buy a company or build it on one's own from the ground up [34]. The formula for business valuation using the replacement method (valuation of infrastructural enterprises—the main asset is infrastructure, e.g., power distribution companies) [8]:

$$W_{ON} = W_{OB} (1 - Z_f)(1 - Z_m),$$

$W_{ON}$—net replacement value (the value of the fixed asset, taking into account its physical and moral wear),
$W_{OB}$—gross replacement value (the value of the new fixed asset),
$Z_f$—physical (technical) wear indicator, $0 \leq Z_f$,
$Z_m$—moral wear indicator (technological change, aging), $Z_m \leq 1$.

A specific case is business valuation using the liquidation method (for the purposes of insolvency proceedings), but then fair value is being dealt with.

Income-based methods consist in estimating the market value of the company understood as the sum of net income obtainable from the company in the future. The general formula of business valuation using the income-based method [10]:

$$WP = \sum_{i=1}^{n} a_t * D_t,$$

t—year of the analysis,
$a_t$—discount rate for the year t,

$D_t$—income in the year t.

In practice, the most frequently applied method is referring valuation to discounted cash flows. The formula for business valuation using the discounted cash flow method [9]:

$$W_d = \sum a_t * NCF_t + RV,$$

$W_d$—income value,
$a_t$—discount rate for the year t,
NCFt—net cash flows for the year t,
RV—residual value.

There are many types of valuations using the DCF (Discounted Cash Flow) method, varying both in the level of detail and the structure of cash flows, as well as the determination of the discount rate. According to the DCF method, the value of the company equals the sum of cash flows discounted at an appropriate rate, which, after cumulation and summing, creates the total cash flow at the disposal of the owners [1].

Asset-based methods, as the oldest concept of the valuation of assets, contemporarily also using the practice of comparable company methods, seem to be to the greatest extent objective and compliant with the intention to determine fair value. On the other hand, income-based methods still require improvement in creating good practices and standards, at least in terms of defining discount rates and the number of years accepted for valuation since, among others, these factors allow for the excessive subjectivity of valuation. Comparable company methods consist in estimating the market value, which is established on the basis of the known sale and purchase transactions. The principle of this method is applied when valuating assets, the prices of which are adjusted to market values. However, the method of comparable company valuation is also the method based on market multiples, which is based on the assumption that the financial market provides the best information for business valuation. The selection of multiples and their use is complicated and causes a lot of controversies, particularly in pursuit of fair value. Therefore, the choice of multiples undoubtedly belongs to the subjective determinants of business valuation.

Mixed valuation methods combine the methods of the valuation of assets with income-based methods. This is due to the assumption that the value of the company is affected not only by its assets but also by the ability to generate income. However, these methods may be unreliable, and this is associated with the possibility of overestimation or underestimation in valuation due to different proportions in the value of assets and profitability in the formulas proposed. Table 6 contains examples of valuation for various mixed methods using the example of actual data presented in the court reports in Katowice and Cracow. The examples were selected because of the same assumptions concerning the valuation using the income-based method. The valuators chose the methods in the following order: the Swiss, German and Stuttgart methods.

**Table 6.** The examples of valuation for various mixed methods.

| Method | Assets and Income | Stuttgart Formula: $W = M + (5r/1 + 5r)(D - M)$ | Anglo-Saxon Formula: $W = M + [1 - 1/(1 + r)^n](D - M)$ | German Formula: $W = (M + D)/2$ | Swiss Formula: $W = (2D + M)/3$ |
|---|---|---|---|---|---|
| Valuation (in PLN Thousand) | $M = 120$ $D = 410$ | 216 | 230 | 265 | 313 |
| | $M = 98$ $D = 546$ | 247 | 268 | 322 | 397 |
| | $M = 290$ $D = 110$ | 230 | 221.5 | 200 | 170 |

Source: Own study based on the valuations by appraisers at the District Court in Katowice and Cracow.

Depending on the subjective choice of the method by the valuator, significant differences in the final valuation may be observed, with low values of the company. Therefore, the objective should be to develop standards that will determine acceptable methods (patterns) for specific industries or cases.

In the above example, it can be seen that the Swiss method prefers (valuates higher) companies that bring higher income with fewer assets, and therefore prefers companies in which assets are less important, e.g., modern IT companies. While the Stuttgart method will price companies higher having a high value of assets with less importance assigned to income possibilities. In this case, it is possible to choose a method that is more favorable to one of the valuation stakeholders of the company. Most often it is a contradictory interest, because one party wants a higher and the other wants a lower valuation. However, as part of the standard and certain norms, objective valuation of the business should be sought: the so-called fair value price. That is why it is so important to define norms and standards even for specific industries. There are also possibilities to create econometric models that, depending on the changing values of determinants, will create a valuation tailored to specific conditions and types of companies. This is an interesting research direction in this field, which requires numerous numerical calculations to develop appropriate patterns. This clearly shows the need for development and the need to come up with new proposals for business valuation methods, which can be seen today.

*4.3. The Methodology of Business Valuation According to the MDI-R Concept*

The contemporary methodology of business valuation according to the authors' concept takes place in accordance with MDI–R (assets, income, intellectual capital and all these components embedded in the market) [35]. Changes in the economic realities of the world (globalization and rapid technological development) have resulted in alterations in valuation methods according to the change in the significance of the components affecting the company's value. Nowadays, the so-called intangible and legal assets are the most important for the total valuation, which has resulted in the natural development and changes in the concepts of business valuation. Additionally, due to the complexity of issues and the share of all historically known components in the total valuation, undoubtedly, one should look for synthetic and universal calculation models. Mixed methods combine asset-based and income-based calculations. The methodology of MDI–R additionally takes into account comparable company methods, as well as the intangible assets and legal valuation. This results from the assumption that the value of the company is affected not only by its assets, including intangible and legal assets, but also by the ability to generate income and the current market (economic) situation of the environment. The company's assets are valuated using the following general formula [36]:

$$M = A - P_o,$$

where:

M—the company's assets,
A—assets,
Po—foreign liabilities,

where the value of fixed assets can be calculated according to the following formula:

$$Wst = Wn(1 - Zf),$$

where:

$W_{st}$—value of the fixed asset,
$W_n$—value of a new fixed asset,
$Z_f$—wear rate of the fixed physical asset, in the range of $0 \le Z_f \le 1$,

or using the market comparable company method, i.e., the current price of the worn fixed asset in the market.

Income is calculated according to the general concept adopted in income-based methods, in particular, using widely known and utilized indicators, for example:

$$D = \frac{NOPAT}{WACC},$$

where:

*NOPAT*—projected annual net operating profit after tax,
*WACC*—discount rate, reflecting weighted average cost of capital of the valuated company.

The DCF method is the most popular income-based method. According to the DCF method, business value equals the sum of discounted cash flows generated by the enterprise, which, after cumulation and summing, create the total cash flow at the disposal of the owners [31].

The MDI–R calculation model takes into account the so-called intangible and legal assets, i.e., human potential, know-how, reputation, the company's market position, etc. These assets are not subjected to the objective valuation since the assets of this type depend on the subjective behavior of market entities, which decide how much they are able to offer for intangible and legal assets. Therefore, these values are considered (valuated) using the comparable company method, consisting in comparisons with transactions of similar companies in the same industry [34]. Depending on the quantity of available transactions, the formula is the following:

$$W_{NiP} = \frac{(W_1 + W_2 + \ldots + W_n)}{n},$$

where:

$W_{NiP}$—market value of single, specific intangible and legal assets,
$w_1, w_2$—known market prices of similar intangible and legal assets,
$n$—number of transactions of the specific value.

The importance of intellectual capital has increased substantially in recent years since business value is less and less dependent on tangible factors. This is caused by global technological, organizational changes, etc., which have led to the knowledge-based economy [35]. Intellectual capital, among others, consists of:

- Specific knowledge, experience, technology,
- Legal assets (brand, know-how, reputation, etc.),
- Relationships with customers and professional skills.

At the same time, the issues associated with intellectual capital are still not well-known and there are many ambiguous and various solutions for both theory and practice in this field [37].

Undoubtedly, due to difficulties in the valuation of the so-called intangible and legal assets, which primarily determine the value of enterprises, in particular highly developed ones, in terms of technology, one deals with difficulties in achieving the so-called fair business valuation. Nowadays, the conclusion is that, in order to make the best possible valuation, it is necessary to valuate individual components affecting the value of the company with separate methods that best reflect the nature of their value. This rule is obeyed by the MDI–R model, which allows the estimation of the total business value, i.e., takes into account all the components affecting business value while assuming that human capital (knowledge, skills) remains unchanged. Business valuation, in accordance with the MDI–R concept, will present the overall image of changes in the company's value, as well as the ones resulting from changes in the business environment. Such valuation will be a good measure of the effectiveness of operations conducted in the enterprise. The characteristics of the MDI–R methodology include:

- Taking into account both tangible and intangible assets (WNiP),

- Taking into account the abilities to generate income,
- Taking into account the current situation in the market in which the company operates,
- Taking into account the company's financing method,
- Taking into account future investment needs,
- Universality—variability of the applied methods in the valuation of specific components, taking into account the conditions for the specific industry.

Therefore, it can be concluded that the MDI–R methodology presents the overall image of changes in the value of the company, as well as the ones that result from changes in the business environment and have an impact on its value. The vast amount of frequently contradictory information concerning economic processes is a characteristic feature of the contemporary knowledge-based economy. Therefore, it is very important to include intangible assets in the valuation, which is an extremely complicated challenge.

The econometric model using an equation (system of equations) helps to explain the mechanism of changes occurring in the studied area. It describes the relationships between given economic quantities. This is a formal mathematical record of existing economic regularities.

$$Y_i = \alpha_0 + \alpha_1 X_{i1} + \alpha_2 X_{i2} + \varepsilon_i.$$

A basic example of a linear model is also used in the valuation model, although the actual relationship can be more complicated, which evolves through a number of practical studies. Building an econometric model requires not only good knowledge of economic theory and mathematical and economic knowledge, but also knowledge of economic practice. The econometric model should not only have a cognitive value from the point of view of economic theory, but also a practical value, which means that it can serve as a tool of inference in the future.

In the valuation model, the explained value is the business value. The main explanatory variables are the value of company assets and the value of income brought. The random component contains, among others, all variables omitted or errors in measurements and calculations. This property is closer to real variables, to what is happening in the real world. After all, we can never determine perfectly straightforward functional dependencies. The function of such a component is performed by the comparative method, which is used to calculate the explanatory variables $M_R$ and $D_R$. That is why the comparative method often appears as a peculiar random component in the econometric model regarding the valuation of enterprises. It allows comparison with transactions (of individual components and also of the most similar companies) that have already taken place on the market. Such a synthetic and universal calculation model is a way to eliminate many risks and problems in business valuation that need to be faced in the future.

As part of the available valuations made by appraisers in the District Court in Katowice and Krakow, a linear regression was performed to estimate structural parameters in the valuation model based on the MDI-R methodology. The companies valued belong to the IT industry. The first row in Table 7 contains estimates of structural parameters, the second row contains standard errors of these estimates. The coefficient of determination (determination) informs about how much of the variability (variance) of the variable explained in the sample coincides with the correlations with the variables contained in the model. It is therefore a measure of the extent to which the model matches the sample. The bigger the regression line, the better suited the data. If the value is between 80% and 90%, then the match is considered good, as in the following example.

**Table 7.** Linear regression model in MDI-R methodology for known valuations at the District Court in Katowice and Krakow for the IT industry.

| $\alpha_0$ | $\alpha_1$ | $\alpha_2$ |
|---|---|---|
| −0.864 | 0.379 | 0.621 |
| S ($\alpha_0$) | S ($\alpha_1$) | S ($\alpha_2$) |
| 4.791 | 0.119 | 0.252 |
| | $R^2 = 0.8065 = 80.65\%$ | |

Source: own study.

$$M_R = (W_{NiP} + A) - P_o,$$

$$D_R = \sum_{i=1}^{n} a_t * NCF_t + RV,$$

$$WP = 0.379 * M_R + 0.621 * D_R - 0.864.$$

Undoubtedly, structural variables require further practical research. For example, the search for more complex relationships than the linear relationship and the values of the structural variables themselves should be assumed. Nowadays, due to the knowledge-based economy (the greater significance of intangible assets and the smaller value of tangible assets), income value is more important in valuation. This allows for practical calculations in the proposed model, as shown in Table 8. However, there are still and will continue to be industries where the high value of material assets will be necessary for business operation. Therefore, estimating structural variables for these differences will require different equations. This will allow a search for the optimal method of business valuation. Currently, it can be concluded that due to differences in industries (resulting from differences in the value of owned assets), a system of equations will most likely be required that will allow tailoring of the selected equation to a particular industry or type of company.

**Table 8.** The examples of valuation for MDI-R methods.

| Method | Assets and Income | MDI-R |
|---|---|---|
| **Valuation (in PLN thousand)** | M = 120 <br> D = 410 | 299 |
| | M = 98 <br> D = 546 | 375 |
| | M = 290 <br> D = 110 | 177 |

Source: own study.

Summing up the business valuation methods, it can be concluded that the appropriate model for the valuation of the economic entity should not only inform about the total value but also indicate the structure of the sources of its creation. Therefore, business valuation methods should take into account as many components as possible of the company affecting its value, which was proposed in the MDI-R concept. Additionally, the proper valuation should be adjusted to the nature of the company and the specific areas of its business activity.

## 5. Discussion and Conclusions

Within the framework of this article, the essence and manner of business valuation using asset-based methods, which constitute the basic and historically first approach to the monetary

business valuation, were determined. Within the framework of the conducted analysis, in terms of the new approach, the reasons for the contemporary crisis of confidence in the business valuation process were presented, including problems and risks occurring in asset-based methods, which, according to the subject literature, are the oldest and most objective foundations for the valuation of a company's assets. While aimed at solving the existing problems and risks, the possibilities of implementing new rules that will restore the quality and confidence in standards for determining the monetary value of a company were determined. Within the framework of determining the direction towards the accomplishment of the objective, which is the most objective valuation of enterprises, the authors' proposals of changes were depicted, which at least related to the need to specify certain valuation methods that can be used in relation to specific industries of the economy and categories of components creating the valuated assets. Nowadays, intangible assets and the ways of their use are more important than the category of so-called material substances, which determine the abilities of the company to multiply the invested capital (including, among others, human potential, brand, know-how, etc.). In the study, through the prism of the analysis and classification of asset-based business valuation methods used in practice, it was shown that the adjusted net assets valuation method is currently the best (objective and the closest to the model fair value) solution, which is confirmed by the fact that it is the basis for all valuations in economic practice. On the other hand, within the framework of the accomplishment of the authors' objective, through the analysis of the literature and practical examples, the need for improvement in the code of conduct and valuation standards was indicated, both in the overall process and within the framework of individual principles applied under asset-based methods, since there is a serious problem in the application of these methods to intangible and legal assets. At the same time, in the study, the factors that can distort the establishment of fair value were presented, while attention was paid to the multidimensionality and complexity of issues of valuation, due to which the pursuit of optimal solutions requires a number of studies of both scientists and practitioners of the discussed subject matter. The research methodology in this study was based on reasoning regarding the methodology of asset-based methods of business valuation and the analysis of practical examples in order to verify the hypothesis of the existence of critical components of valuation, which enable the use of wide subjectivity in estimating the value of assets. The process of valuation of the company, its specificity and essence pose many problems and controversies. This is connected with the essence of monetary valuation, which is a subjective measure. Then, there are various conditions and numerous procedures and methods for business valuation, presented in the study. At the same time, the reasons for the contemporary crisis of confidence in the methodology of business valuation were presented and the directions of future development were indicated, which are of fundamental importance for the operations and opportunities of the conducted business. The study is a part of the discussion concerning the future and development of subsequent solutions aimed at determining, as far as possible, the most objective patterns and standards of business valuation implemented in the practice of economic life.

Summing up, it can be concluded that the appropriate model for the valuation of the economic entity should not only inform about the total value, but also indicate the structure of the sources of its creation. Therefore, business valuation methods should take into account as many components of the company affecting its value as possible. Nowadays, the valuation process is already moving in this direction, the example of which is a simultaneous use of, most frequently, asset-based, income-based and comparable company methods to determine the final value of the company. The use of several methods of valuation in the course of the applied procedure provides an opportunity to make rational decisions as to the final value of the enterprise. Such a tool will lead to the desired harmonization of the methodology for estimation of the basic parameters serving valuation with such a degree of quality, which will provide the recipients of this information with a selection of accurate decision-making options in the future.

In accordance with the literature on the subject, it was found necessary to search for more perfect methods for optimal valuations of fair value for enterprises. An undoubted contribution is the authors'

proposed hybrid method MDI-R, which draws from existing solutions to improve their functionality and applicability. In addition, the article indicates the possible directions of development of valuation methods and the use of existing valuation models to identify companies threatened with bankruptcy. However, in the case of valuation, it is much more complicated and probably should determine the system of equations that will take into account various industries and economic situations.

**Author Contributions:** Conceptualization, M.K and I.M.; methodology, P.S. and I.M.; software, P.S. and I.M.; validation, M.K. and I.M.; formal analysis, M.K. and I.M.; investigation, P.S.; resources, M.K., P.S. and I.M.; data curation, M.K. and I.M.; writing—original draft preparation, M.K. and I.M.; writing—review and editing, P.S.; visualization, P.S. and I.M.; supervision, P.S. and I.M.; project administration, M.K.; funding acquisition, P.S. All authors have read and agreed to the published version of the manuscript.

**Funding:** The project was financed within the framework of the program of the Minister of Science and Higher Education in Poland under the name "Regional Excellence Initiative" in the years 2019–2022, project number 001/RID/2018/19, the amount of financing PLN 10,684,000.00.

**Acknowledgments:** Many thanks to Leon Dorozik for scientific support and Janina Miciuła for life support.

**Conflicts of Interest:** The authors declare no conflict of interest.

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
