# Peer review of "Modern Methods of Business Valuation—Case Study and New Concepts"

_sustainability, doi:10.3390/su12072699_

Round 1
Reviewer 1 Report
I recommend first of all arranging the bibliographic sources in alphabethical order.
Secondly, I recommend completing the conclusions by identifying the most relevant method of evaluating the business based on the data used in the practical exemple.
Finally, I recommend using an econometric model to highlight an optimal method of evaluating the business.
Author Response
Thank you very much for reviewing and pointing out corrections that will improve the article.
As recommended, the article extends the conclusions on the valuation methods based on the data used in the practical example. This confirms the need for further search for enterprise valuation norms and standards, and the use of the econometric model recommended by the reviewer is a good idea, which is confirmed by other studies. This will allow for greater flexibility and adjustment of variable parameters, but this requires extensive quantitative research to develop appropriate values. That is why it is an interesting proposition for future research that is necessary, which is also shown in the discussed article.
As for the bibliography recommendation, arranging bibliographic sources in order of citation is a formal requirement of the publishing house.
Once again, thank you for the valuable review and extension of the authors' view on the usefulness of econometric modeling, as is the case when assessing enterprises at risk of bankruptcy.
Reviewer 2 Report
This study reports on „Modern methods of business valuation - case study and new concepts”. I am pleased that the authors have chosen this topic. This is an interesting and timely focus, which is high on the agenda. In the modern world, the term enterprise value and valuation are of great importance. The written style, logic structure, the introduction, and methodology provide a useful explanation for the results or reach a solid conclusion.
I ask that the authors specifically address each of my comments in their response. I hope my comments, observations, and suggestions will allow the authors to improve the manuscript and work towards publication. Below, I include comments pointing toward some of the issues.
Originality
Research contribution in the paper can be identified; this study can be justified as innovative.
Title
The title is correct as it reflects the objective of the work.
Abstract
The abstract provides a structured summary including contextual background, and result but mission part are the conclusion, and implications of key findings, etc.
Introduction
These section is very well based.
The information provided in the Introduction section is not adequate: What’s the research objective of this paper? What method is adopted? What papers/studies are selected in your review? What are the selection criteria (Year of publication? Quality? Journal? Region?)? And so on. I found this part in the literature review part.
Literature review
The most relevant part is the introduction section that gives a perfect context for the justification of the research. This section includes many relevant references and authors provide theoretical foundations for the analysis using almost appropriate references. The subject is worthy investigation and a lot of information presented in the manuscript is new.
The objective of the article is to demonstrate the need to improve the code of conduct and valuation standards. Within the framework of the accomplishment of the objective, the multidimensionality, as well as complex issues of valuation, have been presented along with the factors that could distort the establishment of fair value.
Methodology
Please prepare methodology part and restructure the manuscript.
The methodology of the study is based on inferences based on the methodology of business valuation and verification on practical examples on the basis of which the hypothesis on the existence of critical elements of valuation has been verified, which allows the use of broad subjectivity in estimating the value of 2assets.
Discussion and Conclusion
The discussion section should make clear the paper’s added value to the existing knowledge and point out the theoretical contributions. The conclusion section should highlight more clearly how the results compare with the recent literature review.
English language and style: moderate English changes required.
Recommendation: minor revision
There are concerns with respect to the presentation of the results of the manuscript as noted above. Addressing these concerns would entail revisions to this manuscript in order to be considered for publication. On the basis of these observations, this paper is recommended for publication after the revisions detailed in my comments are satisfactorily completed.
Author Response
Thank you very much for the review and for suggestions and required corrections that will improve the article.
In accordance with the reviewer's recommendation, the manuscript was restructured and a part describing the methodology used in the article was added. Also in the discussion section the explanation regarding the added value of the article to the existing knowledge was expanded and the theoretical contribution was indicated. As a part of practical examples, the conclusions from calculations have been expanded, which also confirm the need for further search of enterprise valuation norms and standards. The summary section highlights the contribution of this article and indicates further research and application opportunities in this topic. The missing elements were also supplemented in the abstract article.
Once again, thank you for the valuable review that increases the quality of the article and extends the scientific view of the authors.
Round 2
Reviewer 1 Report
A first recommendation concerns the alphabetical order of bibliographic landmarks. It is a rather formal recommendation.
The most important recommendation concerns the use of an econometric model in terms of testing one of the evaluation methods mentioned in the article. Starting from a database provided within a mediun - term industry for example, the autors can test an econometric software applied to the model of a modern business evaluation method.
This represents the recommendation but also the request that I send to the autors in order to publish this article.
Author Response
Thank you very much for substantive comments that will improve the quality of the article. In the attached article revised in line with comments.
Round 3
Reviewer 1 Report
I find that paper does not contain an econometric model tested on certain databases.
Therefore it was not revised according to the previous request.
I propose a recovery period of this case study.
Author Response
Attached is a revised article as recommended by reviewers.
As part of the available valuations made by appraisers in the District Court in Katowice and Krakow, a linear regression was performed to estimate structural parameters in the valuation model based on the MDI-R methodology. The companies valued belong to the IT industry.
If I acquire a large database of valuations performed in a specific industry, I will determine the structural parameters for the valuation model. It is also good to check several specific industries or types of enterprises, which will give interesting extensive research. Now, the only obstacle is the possibility of obtaining a series of data on the valuations carried out companies.
Thank you for the review and pointing out a very interesting possibility of further scientific research.
Round 4
Reviewer 1 Report
Although we expect an econometric model to be tested with data bases over a certain period of time, I find that this request has been solved in principle.
I am obliged to accept the publication of this article but I still recommend the autors to use econometric instruments and to be seriously tested for future publications in scientific journals unanimously recognized worldwide and indexed Web of Science in particular.